# Skin Barrier Function and Infant Tidal Flow-Volume Loops—A Population-Based Observational Study

**DOI:** 10.3390/children10010088

**Published:** 2022-12-31

**Authors:** Martin Färdig, Hrefna Katrín Gudmundsdóttir, Angela Hoyer, Karen Eline Stensby Bains, Catarina Almqvist, Christine Monceyron Jonassen, Eva Maria Rehbinder, Håvard O. Skjerven, Anne Cathrine Staff, Riyas Vettukattil, Cilla Söderhäll, Karin C. Lødrup Carlsen, Björn Nordlund

**Affiliations:** 1Department of Women’s and Children’s Health, Karolinska Institutet, 171 77 Stockholm, Sweden; 2Astrid Lindgren Children’s Hospital, Karolinska University Hospital, 171 64 Stockholm, Sweden; 3Division of Paediatric and Adolescent Medicine, Oslo University Hospital, 0424 Oslo, Norway; 4Institute of Clinical Medicine, Faculty of Medicine, University of Oslo, 0318 Oslo, Norway; 5Department of Medical Epidemiology and Biostatistics, Karolinska Institutet, 171 77 Stockholm, Sweden; 6Department of Virology, Norwegian Institute of Public Health, 0213 Oslo, Norway; 7Genetic Unit, Centre for Laboratory Medicine, Østfold Hospital Trust, 1714 Kalnes, Norway; 8Department of Dermatology and Vaenerology, Oslo University Hospital, 0424 Oslo, Norway; 9Division of Obstetrics and Gynaecology, Oslo University Hospital, 0424 Oslo, Norway

**Keywords:** eczema, filaggrin, infant, PreventADALL, respiratory function tests, transepidermal water loss

## Abstract

Background: The relationship between the skin barrier- and lung function in infancy is largely unexplored. We aimed to explore if reduced skin barrier function by high transepidermal water loss (TEWL), or manifestations of eczema or Filaggrin (*FLG*) mutations, were associated with lower lung function in three-month-old infants. Methods: From the population-based PreventADALL cohort, 899 infants with lung function measurements and information on either TEWL, eczema at three months of age and/or *FLG* mutations were included. Lower lung function by tidal flow-volume loops was defined as a ratio of time to peak tidal expiratory flow to expiratory time (*t*_PTEF_/*t*_E_) <0.25 and a *t*_PTEF_ <0.17 s (<25th percentile). A high TEWL >8.83 g/m^2^/h (>75th percentile) denoted reduced skin barrier function, and DNA was genotyped for *FLG* mutations (R501X, 2282del4 and R2447X). Results: Neither a high TEWL, nor eczema or *FLG* mutations, were associated with a lower *t*_PTEF_/*t*_E_. While a high TEWL was associated with a lower *t*_PTEF_; adjusted OR (95% CI) 1.61 (1.08, 2.42), the presence of eczema or *FLG* mutations were not. Conclusions: Overall, a high TEWL, eczema or *FLG* mutations were not associated with lower lung function in healthy three-month-old infants. However, an inverse association between high TEWL and *t*_PTEF_ was observed, indicating a possible link between the skin barrier- and lung function in early infancy.

## 1. Introduction

Asthma is a chronic obstructive inflammatory airway disease that often develops early in life [1,2], and affects around 10–15% of school children [3,4]. Childhood asthma is complex and heterogeneous, characterized by reversible airflow obstruction and inflammation triggered by allergens, environmental pollutants and infections [3,5]. The treatment often consist of inhaled bronchodilators for symptom relief, anti-inflammatory drugs such as corticosteroids [3,6] and in serious disease the addition of immunomodulatory medication for control [7,8]. The aetiology is multifactorial and partly still unknown [9,10], while it is likely that the disease origins from early life [11,12]. Atopic dermatitis (AD) in early infancy is a risk factor for, and often precedes asthma as well as other allergic diseases [13,14]. Collectively, the progression from one allergic disease to another is often referred to as the atopic march, indicating a link between the allergic diseases [15,16].

Epithelial barrier dysfunction may be involved in allergic disease development, including the airways and the skin [17,18]. The skin acts as a natural protective barrier from allergens and other environmental exposures [19,20]. In AD, an inflammatory condition characterized by eczematous itchy skin lesions [15,21], reduced skin barrier function and dysregulation of the immune system play an important role [22,23]. Increased transepidermal water loss (TEWL), a non-invasive measurement of skin barrier function, has been associated with development of allergic sensitization [24,25] and AD [26,27], and has been shown to precede the clinical manifestations of AD [26,28]. Reduced skin barrier function may also play a role in the development and progression of asthma [29,30]. By the activation of T cells due to cutaneous allergens on a permeable skin barrier [28], accelerated by loss-of-function mutations in the Filaggrin (*FLG*) gene [29], prenatal and early postnatal exposures may have a lifelong effect on respiratory health [3]. *FLG* mutations are linked to higher TEWL [31], eczema [31,32], AD [33,34], allergic sensitization and asthma severity [29,35,36].

Lower lung function, determined by tidal breath flow-volume (TFV) loops, is associated with higher risk of developing asthma in childhood [37], and reduced lung function from birth to adolescence has been observed in children with allergic asthma and AD [38]. Compared to healthy children, infants and children with obstructive airway diseases generally reach peak tidal expiratory flow (*t*_PTEF_) earlier in the expiratory phase with a subsequent lower ratio of time to peak tidal expiratory flow to expiratory time (*t*_PTEF_/*t*_E_) [39,40,41].

Though reduced skin barrier function is strongly associated with AD, and AD as well as lower infant lung function are known risk factors for asthma, the association between the skin barrier- and lung function in early life is largely unexplored. We hypothesized that a reduced skin barrier function could be a common origin of AD and lower lung function in early life. Our primary aim was to explore if high TEWL, and secondarily manifestations of eczema or *FLG* mutations were associated with lower lung function in healthy three-month-old infants.

## 2. Materials and Methods

### 2.1. Study Design and Study Population

The present study is a prospective observational study, using information from the Preventing Atopic Dermatitis and ALLergies in children (PreventADALL) cohort. The study population consisted of 899 participating infants with available TFV loop measurements in the awake state, and information on either TEWL, eczema at three months of age and/or *FLG* mutations. The PreventADALL study, consisting of 2394 mother-child pairs actively participating, is a Scandinavian multicentre population-based prospective birth cohort study with two factorial designed randomized controlled interventions; skin emollients from two weeks to nine months and food introduction from three months of age [42]. Pregnant women were recruited between 2014 and 2016 during the routine ultrasound examination at 18 weeks gestational age (GA) in Norway and Sweden. The participating infants, born without serious illnesses at GA ≥35 weeks, were enrolled during the first days of life.

Informed written consent was collected from the mothers at enrolment, and from both parents at infant inclusion. The PreventADALL study, including the present sub study, was approved by the Regional Committee for Medical and Health Research Ethics in South-Eastern Norway (2014/518) and in Sweden (2014/2242-31/4), as well as registered at clinicaltrials.gov (NCT02449850). The data is stored at the project database in “Service for Sensitive Data” (TSD) at Oslo University, in compliance with General Data Protection Regulation (GDPR) legislation.

### 2.2. Data Collection

#### 2.2.1. Three-Month Clinical Investigation

Lung function by TFV loops was measured in calm awake infants positioned in a semi-recumbent position. The TFV loop parameters were sampled and recorded using the employed Exhalyzer^®^ D (Eco Medics AG, Duernten, Switzerland), with a soft-rimmed air-inflated face mask covering the mouth and nose. All measurements were manually scrutinized with focus on loop shape and reproducibility by one of three trained investigators in Oslo (Norway) and in Stockholm (Sweden). Technically unacceptable loops were manually removed before approval. Further information about the standard operating procedure for TFV loop measurements and manual loop selection is described elsewhere [43].

TEWL (g/m^2^/h) was measured on the left lateral upper arm with an open chamber DermaLab USB (Cortex, Hadsund, Denmark), at room temperature between 20 °C to 25 °C, in line with international recommendations. All ranges of humidity from 6.40% to 72.9%, with a mean of 30.9%, were accepted in line with previous findings [44]. After 15 min of acclimatization, with the infants wearing only diapers, three consecutive measurements were performed in calm infants, distanced from direct sunlight, while windows and doors were kept closed.

Clinical skin examinations were performed by trained study personnel, educated at workshops to minimize inter-observer variability. Parents were advised not to bathe the infants or use any skin emollients for at least 24 h prior to the visits. Observation of eczematous skin lesions suggestive of AD were verified by a medical doctor, clinically excluding common differential diagnoses to AD [45].

#### 2.2.2. FLG Mutations

DNA was isolated from blood and genotyped for the most common *FLG* mutations in Europeans; R501X, 2282del4 and R2447X, using the TaqMan-based allelic discrimination assay, as previously described by Hoyer et al. [32]. Allele-specific Taqman MGB (minor groove binder) probes were labelled with the fluorescent dyes FAM and VIC, respectively. Polymerase chain reactions (PCR) were carried out in 384-well plates with use of genomic DNA, a reaction mix containing the specific TaqMan assay solution and 1X TaqMan Universal PCR Master Mix (Applied Biosystems, Foster City, CA, USA). Amplification was done following the Taqman Universal PCR protocol. Allelic discrimination was performed with the QuantStudion 6 and 7 Flex (QS6 and 7 FLX).

#### 2.2.3. Birth and Background Characteristics

Background data were collected from electronic questionnaires answered at approximately 18 and 34 weeks of pregnancy and three months postpartum. Birth data, including anthropometrics and health were recorded from electronic medical records. At the three-month clinical investigation, infant weight and length were measured according to a standard operating procedure.

### 2.3. Definitions

#### 2.3.1. Primary Outcome

Lower *t*_PTEF_/*t*_E_: a *t*_PTEF_/*t*_E_ <0.25, previously associated with airway obstruction and asthma [37,46,47,48].

#### 2.3.2. Secondary Outcome

Lower *t*_PTEF:_ a *t*_PTEF_ <0.17 s (below the 25th percentile) was chosen, as lower values have previously found in infants and children with airway obstruction and asthma [39,40,41].

#### 2.3.3. Sensitivity Analysis

The continuous *t*_PTEF_/*t*_E_ and *t*_PTEF_.

#### 2.3.4. Exposures

High TEWL: a mean TEWL value >8.83 g/m^2^/h (above the 75th percentile), in line with previous studies [24,44].

Eczema: Defined as clinically observed eczematous skin lesions with the exclusion of common differential diagnosis to AD such as infantile seborrheic dermatitis and irritative contact dermatitis [45]. As few three-month-old infants fulfil the strict diagnostic criteria for AD by the United Kingdom Working Party (UKWP) [49], eczema was used as a proxy.

*FLG* mutations: Being carrier of any of the three mutations R501X, 2282del4 and R2447X of the *FLG* gene, hypothesized to contribute to asthma [15].

### 2.4. Statistical Analyses

Comparisons were performed using parametric or non-parametric tests; independent *t*-tests for continuous variables presented with means, standard deviations (SD) and minimum-maximum (min-max), and Chi^2^ tests for categorical variables presented with numbers (n) and percentages (%). The TFV loop parameters are presented with mean (SD; min-max) for continuous variables. Associations between high TEWL, eczema, *FLG* mutations and lower *t*_PTEF_/*t*_E_ and *t*_PTEF_ were examined using univariate and multivariate logistic regression models, presented with odds ratios (OR) and 95% confidence intervals (CI). In a sensitivity analysis, the univariate and multivariate linear associations between high TEWL, eczema and *FLG* mutations, and the continuous *t*_PTEF_/*t*_E_ and *t*_PTEF_ were explored, presented with ß coefficients (95% CI). Analyses for possible interactions between the exposures (high TEWL, eczema or *FLG* mutations) and the skin intervention, and the primary and secondary outcomes (lower *t*_PTEF_/*t*_E_ and *t*_PTEF_) were conducted. The statistical significance for all tests was set to 0.05. Statistical analyses were conducted using IBM SPSS Statistics 26 software.

## 3. Results

### 3.1. Study Population

The study population consisted of 899 infants participating in the PreventADALL study with available TFV loop measurements in the awake state, and information on either TEWL, eczema at three months of age and/or *FLG* mutations. The remaining 1495 infants with missing information on either TFV loop measurements in the awake state, TEWL, eczema at three months of age and/or FLG mutations were excluded (Figure 1). 

The 899 included infants were born at a mean ± SD GA of 40.1 ± 1.32 weeks and 439 (48.8%) were girls. Compared to the remaining cohort not included in this study, the included infants more often had parents with higher socioeconomic status, a lower mean TEWL and higher frequency of eczema, though similar rates of *FLG* mutation carriers. At three months of age, the mean TEWL among 827/899 (92.0%) included infants was 7.94 ± 5.66 g/m^2^/h, eczema was present in 135/898 (15.0%) and *FLG* mutations were identified in 73/730 (10.0%) infants (Table 1), evenly distributed between boys and girls (data not shown). The included infants had a mean ± SD *t*_PTEF_/*t*_E_ of 0.39 ± 0.08. The infants with a lower *t*_PTEF_/*t*_E_ (n = 48) had a mean *t*_PTEF_/*t*_E_ of 0.23 ± 0.02 and a mean *t*_PTEF_ of 0.16 ± 0.04. The infants with a lower *t*_PTEF_ (n = 223) had a mean *t*_PTEF_/*t*_E_ of 0.34 ± 0.08 and a mean *t*_PTEF_ of 0.15 ± 0.01. The TFV loop parameters are further described in Appendix A. 

### 3.2. Primary Analysis: High TEWL, Eczema, FLG Mutations and Lower t_PTEF_/t_E_

A high TEWL was not significantly associated with a lower *t*_PTEF_/*t*_E_ in the multivariate analyses; adjusted OR (95% CI) 1.21 (0.58, 2.53). Likewise, the presence of eczema or *FLG* mutations were not associated with a lower *t*_PTEF_/*t*_E_; adjusted OR (95% CI) 1.09 (0.44, 2.70) and 0.93 (0.27, 3.18), respectively (Table 2, Table 2.1, Figure 2a). The relationship between mean TEWL and the continuous *t*_PTEF_/*t*_E_ at three months of age is displayed in a scatter plot, see Appendix A.

### 3.3. Secondary Analysis: High TEWL, Eczema, FLG Mutations and Lower t_PTEF_

A high TEWL was significantly associated with a lower *t*_PTEF_ in the multivariate analysis; adjusted OR (95% CI) 1.61 (1.08, 2.42). Neither the presence of eczema, nor *FLG* mutations were associated with a lower *t*_PTEF_; adjusted OR (95% CI) 1.19 (0.74, 1.92) and 1.09 (0.57, 2.06), respectively (Table 2, Table 2.2, Figure 2b). The relationship between mean TEWL and the continuous *t*_PTEF_ at three months of age is displayed in a scatterplot, see Appendix A.

### 3.4. Sensitivity Analysis: High TEWL, Eczema, FLG Mutations and Continuous t_PTEF_/t_E_ and t_PTEF_ as Well as Interaction Effects

Neither a high TEWL, nor eczema or *FLG* mutations were associated with the continuous *t*_PTEF_/*t*_E_ in the multivariate analyses (Appendix A). While a high TEWL was negatively associated with the continuous *t*_PTEF_ in the multivariate analysis, the presence of eczema or *FLG* mutations were not (Appendix A).

No significant interaction effects of high TEWL, eczema or *FLG* mutations and the skin intervention on lower *t*_PTEF_/*t*_E_ or *t*_PTEF_ were found (Appendix A).

## 4. Discussion

In this population-based birth cohort study of 899 three-month-old healthy infants, no associations between high TEWL, eczema nor *FLG* mutations, and a lower *t*_PTEF_/*t*_E_ were observed. While a high TEWL was inversely associated with *t*_PTEF_, the presence of eczema or *FLG* mutations were not. The significant associations indicate a possible link between the skin barrier- and lung function in early infancy.

### 4.1. TEWL and Lung Function

This study provides novel insight into a potential link between the skin barrier function and infant lung function. While a high TEWL was not associated with the ratio *t*_PTEF_/*t*_E_ at three months of age, a significant negative association was observed between high TEWL and lower *t*_PTEF_ as well as the continuous *t*_PTEF_. To our knowledge, this is the first study to explore the association between the skin barrier function and lung function within a large population of healthy infants. In relation to asthma in adolescents and adults, a previous study of 95 individuals found no differences in TEWL between allergic asthmatics and healthy controls [50]. Infants and children with obstructive airway diseases generally reach *t*_PTEF_ earlier in the expiratory phase with a subsequent lower *t*_PTEF_/*t*_E_ ratio, compared to healthy children [39,40,41], providing an important rationale for including *t*_PTEF_ as an outcome in this study. Although reduced skin barrier function was not associated with lower lung function measured by *t*_PTEF_/*t*_E,_ the inverse association with *t*_PTEF_, where high TEWL increased the risk of having shorter time before reaching peak expiratory flow, suggests that lung function and skin barrier function may share a common developmental origin, be manifestations of barrier dysfunction observed in two different organ systems, or infer a causal link between the two. However, as no crude associations to *t*_PTEF_ were seen, further longitudinal studies are needed to exclude the possibility that the significant associations represented chance findings. Whether high TEWL in infancy increases the risk of childhood asthma needs to be investigated in future studies.

### 4.2. Eczema and Lung Function

No evidence of any associations between eczema and lower infant lung function was found in this study. As AD is related to asthma development [15], we had hypothesized that early life eczema might also be related to altered lung function. However, though based on older children, our results correspond well with Hu et al.’s findings of no differences in lung function by spirometry and distinctive eczema phenotypes among 4227 school children [51]. Likewise, in 135 children from the Norwegian Environment and Childhood Asthma (ECA) study, lung function development by TFV loops from birth to two years appeared independent of AD [52]. Since few infants fulfil the major requirement of itch at three months of age, the appropriateness of the commonly used UKWP criteria for AD in early infancy has previously been questioned [49]. As only 1.60% of the infants fulfilled the criteria of the UKWP for AD [53] in this study, clinically observed eczema (15.0%) by medical doctors was used as a proxy. Though a link between AD severity and asthma previously have been suggested [54], the severity of eczema was not evaluated in this study. In a previous paper by Lødrup Carlsen et al., reduced lung function from birth to adolescence was retrospectively observed in children with allergic asthma and AD [38].

### 4.3. FLG Mutations and Lung Function

In our study, no associations between *FLG* mutations and lower infant lung function were observed. Among markers associated with allergic diseases, mutations in the *FLG* gene, a protein essential for the skin barrier function [29,30], are hypothesized to contribute to asthma [15]. To our knowledge, there are no previously published reports on *FLG* mutations and infant lung function, and thus, our finding of no association is novel. In relation to symptoms of airway obstruction, the GO-CHILD birth cohort of 2312 British infants discovered an association between *FLG* mutations and wheezing at six months of age [55]. Interestingly, the risk of developing asthma in *FLG* mutation carriers appears to be limited to children with a previous history of AD [30,33].

### 4.4. Strengths and Limitations

There are several strengths to this study. First, infants were enrolled antenatally, and potential risk factors for adjustment in analyses were recorded prior to the clinical investigation from which the present results are presented. The study comprises lung function measurements in almost 900 healthy three-month-old infants from a general population, with TEWL measurements available in 92.0%. The TFV measurements were performed in early infancy, limiting the time for postnatal events to adversely impact lung function development. Both the study personnel measuring TEWL and medical doctors assessing eczema were educated at joint workshops, to increase the inter-observer reliability in both Norway and Sweden. As reported by Hoyer et al., the 9.00% prevalence of *FLG* mutation carriers in the original PreventADALL cohort was comparable to other European studies [32], corresponding with the rates (10.0%) in our study population. The thorough quality assessment of TFV loop measurements performed by the investigators further strengthen our findings [43]. While a *t*_PTEF_/*t*_E_ below 0.25 in infancy has been related to future obstructive lung diseases [37,46,47,48], we are not aware of any defined cut-off values for *t*_PTEF_ in infancy. Infants with obstructive airway diseases generally reach *t*_PTEF_ faster in the expiratory phase compared to healthy children [39,40,41], therefore, as we were interested in the lower range of *t*_PTEF_, a *t*_PTEF_ below the 25th percentile (<0.17 s) was selected as secondary outcome. Furthermore, the continuous *t*_PTEF_ was used in our sensitivity analysis. However, our observations are limited to three-month-old infants which may be a limitation of the study. Repeated observations in the same infants later in infancy might potentially yield more conclusive results.

In addition to higher frequency of eczema and lower mean TEWL, socioeconomical differences between the study population and the excluded infants were found. Some of the differences seen may be explained by the exclusion of infants living in the less densely populated county Østfold (Norway), where lung function was not measured. At inclusion, the approximately 340 infants from Østfold more often lived in rural environments, had younger parents with higher frequencies of previous pregnancies and lower parental education and income, compared to the infants from the capital cities, Oslo (Norway) and Stockholm (Sweden) [42].

Possible modifications of the skin intervention on the associations between TEWL, eczema or *FLG* mutations and lung function, were assessed in interaction analyses. As we found no significant interaction effects between the skin intervention and the exposures on the outcomes, we included all infants in our analyses, regardless of intervention group allocation. Based on the existing literature, parental asthma [56], parental education level [56], nicotine exposure (snus/smoking) in pregnancy [57], sex [45,58], GA at birth [45,59], weight at three months of age [11,60] and the skin intervention were all treated as possible confounders in multivariate regression models. At the time exposures and outcomes were measured the food intervention had not yet started, hence we adjusted for the skin intervention, only. By isolating the effect of potential effect modifiers from the TFV loop parameters, adjusting for relevant confounders, we were also able to detract the possibility that effect-modifying factors, such as the skin intervention, heredity for asthma or infant sex, had influence on our findings.

### 4.5. Clinical Implications for Future Research

With some exceptions, developmental factors for impaired lung function and asthma from early infancy, childhood and later adulthood are not well understood [61]. Based on the significant inverse associations between high TEWL and *t*_PTEF_, our results suggest a possible link between the skin barrier- and infant lung function. One might thereby speculate that reduced skin barrier may influence lung function in early infancy, contributing as one of the missing pieces of the puzzle of understanding lung function development. As children and adults with obstructive airway diseases generally reach *t*_PTEF_ earlier [39,40,41], and in this study infants with a lower *t*_PTEF_ also had a lower *t*_PTEF_/*t*_E_, it is possible that a lower *t*_PTEF_ might better capture infants with an obstructive breathing pattern. However, as *t*_PTEF_ seldomly have been described in the literature it is debatable whether the parameters are sensitive markers for lung function and asthma. Further studies are necessary to establish if our findings could be of relevance for the identification and prediction of infants and older children at risk of developing asthma.

## 5. Conclusions

While no associations between high TEWL, eczema or *FLG* mutations and lower *t*_PTEF_/*t*_E_ were found in healthy infants at three months of age, we are first to report that a high TEWL inversely was associated with *t*_PTEF_, indicating a possible link between the skin barrier- and lung function in early infancy. Future studies are necessary to determine the relationship between the skin barrier function and lung function development in general, as well as in children at risk of developing asthma. 

## Figures and Tables

**Figure 1 children-10-00088-f001:**
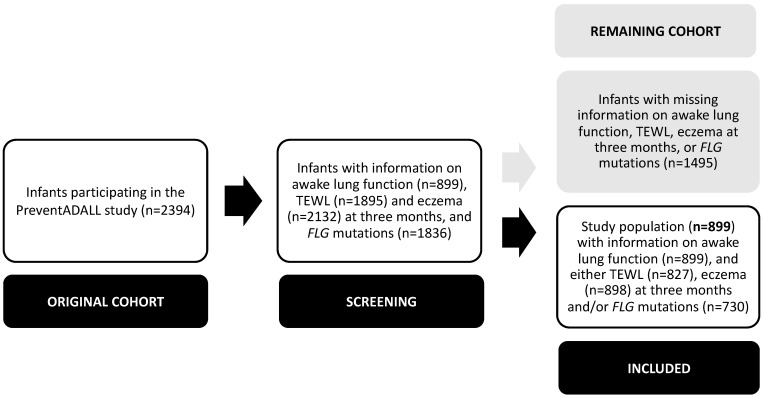
Flow chart of the study population (n = 899). PreventADALL: Preventing Atopic Dermatitis and ALLergies in children; TEWL: transepidermal water loss; *FLG*: Filaggrin.

**Figure 2 children-10-00088-f002:**
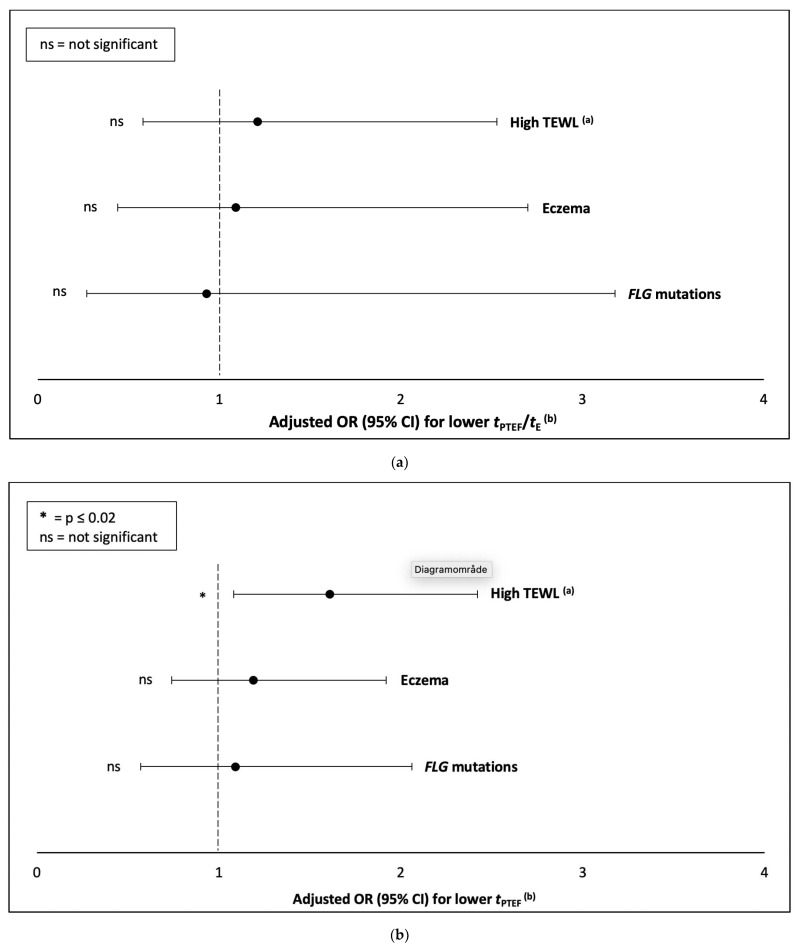
(**a**,**b**). Adjusted OR (95% CI) for lower *t*_PTEF_/*t*_E_ (**a**) and *t*_PTEF_ (**b**), according to the presence of high TEWL, eczema or *FLG* mutations in the 899 three-month-old infants. ^(a)^ High TEWL equal to mean TEWL >8.83 g/m^2^/h (>75th percentile) ^(b)^ Lower *t*_PTEF_/*t*_E_ and *t*_PTEF_ equal to a *t*_PTEF_/*t*_E_ <0.25 and a *t*_PTEF_ <0.17 s (<25th percentile), respectively.

**Table 1 children-10-00088-t001:** Birth and background characteristics of the 899 included infants and the remaining cohort at three months of age.

Characteristics	n (%) or Mean (SD; min–max)	No.	n (%) or Mean (SD; min–max)	No.	*p*-Value ^(a)^
	Included infants (n = 899)		Remaining cohort (n = 1495)		
Parents					
Age mother (n = 2394)	33.0 (3.92; 21–48)	899	32.1 (4.25; 20–47)	1495	<0.001
Age father (n = 2028)	35.2 (5.27; 23–59)	757	34.4 (5.58; 21–72)	1271	0.003
Nordic origin mother (n = 2171)	748 (91.6)	817	1215 (89.7)	1354	0.163
Nordic origin father (n = 2119)	710 (89.5)	793	1193 (90.0)	1326	0.748
High parental education ≥4 years (n = 2117)	612 (76.7)	798	843 (63.9)	1319	<0.001
Low parental income (n = 2134)	67 (8.30)	810	235 (17.7)	1324	<0.001
Urban living environment pregnancy (n = 2171)	793 (97.1)	817	1220 (90.1)	1354	<0.001
Married/cohabitant mother (n = 2179)	799 (97.3)	821	1324 (97.5)	1358	0.801
Parity ≥1 (n = 2391)	330 (36.7)	899	632 (42.4)	1492	0.006
Nicotine exposure in pregnancy (n = 2394)	90 (10.1)	899	163 (11.0)	1495	0.493
Parental asthma (n = 2028)	235 (30.6)	769	372 (29.5)	1259	0.629
Parental atopic dermatitis (n = 1993)	204 (27.0)	756	411 (33.2)	1235	0.003
Infants (birth)					
Girl (n = 2394)	439 (48.8)	899	700 (46.8)	1495	0.340
GA at birth in weeks (n = 2388)	40.1 (1.32; 35.0–42.4)	897	40.0 (1.36; 35.0–42.6)	1491	0.016
Caesarean section (n = 2390)	146 (16.3)	897	251 (16.8)	1493	0.734
Birth weight in g (n = 2384)	3553 (466; 1935–4946)	895	3582 (493; 1794–5632)	1489	0.160
Infants (3 months)					
Age in months (n = 2108)	3.05 (0.24; 1.90–4.50)	889	3.08 (0.28; 2.10–4.90)	1219	0.019
Weight in kg (n = 2126)	6.24 (0.78; 4.40–8.90)	895	6.27 (0.78; 3.70–9.33)	1231	0.401
Length in cm (n = 2099)	61.8 (2.21; 55.5–70.9)	885	61.9 (2.42; 54.0–70.0)	1214	0.809
Weight gain since birth in kg (n = 2117)	2.69 (0.65; 0.99, 4.97)	891	2.69 (0.65; 0.63, 5.26)	1226	0.411
Breastfed exclusively (n = 1852)	542 (69.0)	785	699 (65.5)	1067	0.110
Skin barrier (3 months)					
Mean TEWL (n = 1895)	7.94 (5.66; 0.00–48.5)	827	8.59 (6.06; 1.13–58.0)	1068	0.017
*FLG* mutations (n = 1836)	73 (10.0)	730	93 (8.40)	1106	0.247
Eczema, clinically observed (n = 2132)	135 (15.0)	898	127 (10.3)	1234	<0.001
AD, fulfilling UKWP criteria (n = 2132)	14 (1.60)	898	14 (1.10)	1234	0.390
TFV loop parameters (3 months)					
*t*_PTEF_/*t*_E_ (n = 899)	0.39 (0.08; 0.19–0.63)	899	-	0	na
*t*_PTEF_ (n = 899)	0.21 (0.05; 0.11–0.45)	899	-	0	na

^(a)^ Independent t-test or Chi^2^ test. GA: gestational age; TEWL: transepidermal water loss; *FLG*: Filaggrin; AD: atopic dermatitis; UKWP: United Kingdom Working Party; *t*_PTEF_/*t*_E_: time to peak tidal expiratory flow to total expiratory time; *t*_PTEF_: time to peak tidal expiratory flow; no.: number; na: not applicable.

**Table 2 children-10-00088-t002:** 1,2. Logistic regression models for high TEWL, eczema, or *FLG* mutations and lower *t*_PTEF_/*t*_E_ (2.1) and *t*_PTEF_ (2.2), in the 899 three-month-old infants.

Characteristics	Crude OR (95% CI)	No.	*p*-Value	Adjusted OR (95% CI)	No.	*p*-Value ^(a)^
	Table 2.1. Lower *t*_PTEF_/*t*_E_ ^(b)^					
High TEWL ^(c)^	0.93 (0.46, 1.87)	827	0.842	1.21 (0.58, 2.53)	686	0.611
Eczema	0.96 (0.42, 2.19)	898	0.929	1.09 (0.44, 2.70)	740	0.849
*FLG* mutations	0.72 (0.22, 2.39)	730	0.589	0.93 (0.27, 3.18)	609	0.908
	Table 2.2. Lower *t*_PTEF_ ^(b)^					
High TEWL ^(c)^	1.30 (0.91, 1.86)	827	0.150	1.61 (1.08, 2.42)	686	0.020
Eczema	1.12 (0.74, 1.70)	898	0.593	1.19 (0.74, 1.92)	740	0.475
*FLG* mutations	0.98 (0.56, 1.70)	730	0.932	1.09 (0.57, 2.06)	609	0.796

^(a)^ Logistic regression models adjusted for parental asthma, high parental education (≥4 years of university studies), nicotine exposure in pregnancy, sex, GA, weight at three months of age and the skin intervention. ^(b)^ Lower *t*_PTEF_/*t*_E_ and *t*_PTEF_ equal to a *t*_PTEF_/*t*_E_ <0.25 and a *t*_PTEF_ <0.17 s (<25th percentile), respectively. ^(c)^ High TEWL equal to mean TEWL >8.83 g/m^2^/h (>75th percentile). TEWL: transepidermal water loss; *FLG*: Filaggrin; *t*_PTEF_/*t*_E_: time to peak tidal expiratory flow to expiratory time; *t*_PTEF_: time to peak tidal expiratory flow; no.: number.

## Data Availability

Participants of this study were not asked to consent for open access data from third parties.

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
