# Peer review of "Skin Barrier Function and Infant Tidal Flow-Volume Loops—A Population-Based Observational Study"

_children, 2022, doi:10.3390/children10010088_

Round 1

Reviewer 1 Report

Authors wished to investigate the relationship between reduced skin barrier function, evaluated by  high transepidermal water loss (TEWL), manifestations of eczema or Filaggrin (FLG) mutations, and  lower lung  function, defined  on the basis of the ratio of time to peak tidal expiratory flow to expiratory time (tPTEF/tE) < 0.25 and a tPTEF < 0.17 seconds (<25th percentile)   in three-month-old infants, coming from the population-based PreventADALL cohort. They did not find any association of reduced skin barrier function with lower lung function, but they observed an inverse association between high TEWL and tPTEF.

In the strengths and limitations paragraph Authors report only strengths of their study.

A limitation of the study may be the observation limited at three months-old infants. More conclusive results could come from a repeated observation in the same infants at 6 months of age,

Reviewer 2 Report

INTRODUCTION

- The definition of asthma should be refined and a very short mention to the treatments used at the moment (including biologics) would be also appropriate. Indeed, “Asthma is a heterogeneous chronic inflammatory disease of bronchial airways characterized by reversible airflow obstruction. …the airway inflammation is the main determinant of the airway hyper-responsiveness, which can be triggered by a multitude of agents, including allergens, infections, and pollutants” (see: Omalizumab in the Therapy of Pediatric Asthma. Recent Pat Inflamm Allergy Drug Discov. 2018;12(2):103-109. doi: 10.2174/1872213X12666180430161351). 

- “Epithelial barrier dysfunction may be involved in allergic disease development, including airways and the skin [3]. Eczema in early infancy is a risk factor for, and often precedes asthma as well as other allergic diseases [4].” Here, the authors should clarify that they specifically mean “atopic eczema/dermatitis” (e.g. Genetic/Environmental Contributions and Immune Dysregulation in Children with Atopic Dermatitis. J Asthma Allergy. 2022 Nov 23;15:1681-1700. doi: 10.2147/JAA.S293900). Indeed, as they show later at lines 51-52, eczema is a general dermatological terms and eczematous lesions/eczema may be part of the clinical pictures of other skin disorders that nothing have to do with the risk of (allergic) asthma (e.g. impetigo).

- In general, I would suggest the authors to rearrange the first few paragraphs of the introduction in order to clarify some concepts and provide a better information flow. 

- The final paragraph of the introduction could be shortened and limited to a clear summary of the study general objective and hypothesis, with specific aspects included in the methods section.

METHODS

- Study design and population: the number of study participants is a result, in my opinion. Here, the authors should clearly list and explain inclusion and exclusion criteria. Therefore, also figure 1 could be part of the results, in the end. 

- I also suggest giving a clear and short definition of the study design, first. Then the authors may explain some details and peculiar points. “The present study addressed observational research questions based on information prospectively collected from the Preventing Atopic Dermatitis and ALLergies in children 80 (PreventADALL) study.” Then, is it a prospective observational study, isn’t it?

- I would suggest explaining a little more the definitions, in light of the several references used for this purpose. 

- It seems this study includes genetic information (which is not part of the routine analysis): “FLG mutations: Being carrier of any of the three mutations R501X, 2282del4 and 143 R2447X of the FLG gene.” If so, analytical methods should be included. The authors refer to article [28], but a few details would be useful.

- “CI. Parental asthma 152 [32], parental education (≥4 years of university studies) [32], nicotine exposure (snus/smoking) in pregnancy [33], sex [27,34], gestational age (GA) at birth [27,35], weight 154 at three months of age [2,36] and the skin intervention were all treated as possible confounders.” I am not sure this is part of statistical methods description. It sounds more as a point to be disclosed in the discussion.

- There is no ethical statement subsection included in the main manuscript. Please, add it anyway, even though there is ethical statement at the end, including both IRB references and informed consent information.

RESULTS

- Since this section is correctly organized in subsections, even the first part of the results should be organized so. Consider the previous comments related to figure-1 related information to revise this point. 

- Current subsection 3.1, 3.2 and 3.3 should be expanded in terms of results description. The authors should also highlight the main findings related to the linked tables, including supplementary materials. 

DISCUSSION

- Overall, the discussion sounds well-organized. 

- My only remark, according to the main study conclusion (“an inverse association between high TEWL and tPTEF was observed”) is that the authors may explain better the main relevance of this observation in practical terms in the appropriate discussion subsection. Indeed, the readership is a general pediatric readership and some aspects related to lung function analysis may not be immediately clear to all. 

CONCLUSION

- I think the authors should emphasize more the novelty and practical relevance of the study findings. 

Reviewer 3 Report

The mere idea of seeking the roots of asthma in early childhood is fundamental and fruitful. I think that the parameters chosen are appropriate and important, however, I would suggest incorporating additional parameters of barrier dysfunctions in other compartments, e. g. some occludin, claudin, JAMs, E-cadherins, desmoglein, desmoglein, F-actin, lectins, integrins, zonulin. I think also, that the very early age of subjects may not allow for registering the consequences of these dysfunctions on the respiratory system - usually, the first symptoms are from the skin and the DI tract.

Reviewer 4 Report

Manuscript entitled „Skin barrier function and infant tidal flow-volume loops – a population based observational study"  is an original article. Authors have presented the results of a study where possible relationships between altered skin barier function and lower lung function in three-month-old infants (n=899) were investigated. The study was based on data collected from PreventADALL study.The study was approved by the ethics committee and it was registered at clinicaltrials.gov (NCT02449850). In short, skin barrier function was evaluated by determining transepidermal water loss (TEWL), by DNA genotyping of the most common mutations in the filaggrin gene (FLG) and by clinical skin examination of eczematous skin lesions. Lung function was determined by tidal breath flow-volume (TFV) loops. The results showed that neither high TEWL, nor FLG mutations or clinical features of eczema were not associated with lower lung function. Still, a high TEWL was associated (adjusted OR) with an earlier reach of peak tidal expiratory flow (usually observed in infants or small children with ostructive airway diseases). Authors concluded that lower lung function in subjects was not associated with high TEWL, eczema and FLG mutations. However, high TEWL was inversely associated with elapsed time to peak tidal expiratory flow. Thus, future studies are needed to determine hypothetical link between skin barrier function and lung function. The manuscript is written correctly. The text is accompanied with two figures and two tables. All of them are useful and informative. The discussion is properly written containing enough criticism including strengths and limitations of the study. Citated literature sources (n=43) are relevant and up-to-date. Apart from praise, I have no other comments or suggestions for the authors.

Round 2

Reviewer 2 Report

The authors improved the manuscript. I have no further major revisions to suggest.